# A Comparative Study of Experimental Configurations in Synchrotron Pair Distribution Function

**DOI:** 10.3390/ma12081347

**Published:** 2019-04-25

**Authors:** Jesus D. Zea-Garcia, Angeles G. De la Torre, Miguel A. G. Aranda, Ana Cuesta

**Affiliations:** Departamento de Química Inorgánica, Cristalografía y Mineralogía, Universidad de Málaga, 29071 Málaga, Spain; jdavidzea@uma.es (J.D.Z.-G.); mgd@uma.es (A.G.D.l.T.); migarcia@cells.es (M.A.G.A.)

**Keywords:** synchrotron radiation, total scattering, cement samples, C–S–H gel

## Abstract

The identification and quantification of amorphous components and nanocrystalline phases with very small crystal sizes, smaller than ~3 nm, within samples containing crystalline phases is very challenging. However, this is important as there are several types of systems that contain these matrices: building materials, glass-ceramics, some alloys, etc. The total scattering synchrotron pair distribution function (PDF) can be used to characterize the local atomic order of the nanocrystalline components and to carry out quantitative analyses in complex mixtures. Although the resolution in momentum transfer space has been widely discussed, the resolution in the interatomic distance space has not been discussed to the best of our knowledge. Here, we report synchrotron PDF data collected at three beamlines in different experimental configurations and X-ray detectors. We not only discuss the effect of the resolution in Q-space, Q_max ins_ of the recorded data and Q_max_ of the processed data, but we also discuss the resolution in the interatomic distance (real) space. A thorough study of single-phase crystalline nickel used as standard was carried out. Then, selected cement-related samples including anhydrous tricalcium and dicalcium silicates, and pastes derived from the hydration of tricalcium silicate and ye’elimite with bassanite were analyzed.

## 1. Introduction

Powder diffraction, normally coupled to the Rietveld method, is one of the most used techniques to study the structure and properties of crystalline materials [1]. This approach allows a fast and non-destructive analysis of all component mixtures with an easy sample preparation. Consequently, it is possible to make a quick test of unknown materials and to perform materials characterization in such fields as materials science, chemistry, physics, mineralogy, etc. [2].

An ordinary crystallographic analysis is based on Bragg reflections which are related to the long-range, in most cases periodic, average structure of materials [3]. However, this type of analysis ignores the weak diffuse scattering that can provide information about local atomic structure(s). The total scattering technique is a relatively new alternative methodology which can provide information about atomic local order. That involves the Bragg and diffuse scattering through the analysis, in most cases, of the atomic pair distribution function (PDF). The PDF data are obtained by a Fourier transformation of the total scattering powder diffraction pattern [4,5,6]. This methodology has been successfully applied to the study of amorphous materials without long range order and crystalline materials as well [4,5,6,7,8,9].

The reduced PDF function, also known as G(r), give us the possibility of finding pairs of atoms separated by a r distance. G(r) is obtained from Equation (1) [10]:(1)$$G\left(r\right)=4\pi r\left[\rho \left(r\right)-\rho_{0}\right]=\frac{2}{\pi }\int_{0}^{\infty }Q\left[S\left(Q\right)-1\right]\sin\left(\mathrm{Qr}\right)\mathrm{dQ}$$

Here, *ρ*(r) is the microscopic atomic pair density, *ρ*_0_ is the average atomic number density, S(Q) is the total scattering structure function, Q is the momentum transfer, Q = 4πsin(θ)/λ.

The use of a short wavelength and high 2θ diffracting angles allows to access to a large Q-range which is very important [11] to obtain PDF patterns with (very) high quality. The employed finite Q-range introduces errors in the PDF data which are evident as termination ripples. The amplitudes of these ripples increase as the value of Q_max_ decreases [4,12]. Total scattering data analyzed by PDF methodology allow distinguishing between components that are amorphous, nanocrystalline and/or crystalline by focus the analysis in the appropriate r-windows. In addition to the local bonding environment information, PDF can also provide quantitative information concerning phase contents [4,13,14,15].

It is important in a PDF analysis to take into account the instrumental parameters such as Q_damp_, [4] which is the dampening parameter, and Q_broad_ [16], which is the peak broadening parameter. These parameters are characteristic for each experimental setup and can be determined with a highly crystalline material used as standard. It is important to point out that the Q_damp_ parameter is directly related with the finite Q-resolution through the Gaussian shape of the peak attenuation. Consequently, it is important to collect unbiased diffraction data up to the largest possible Q-value. A comparison of the resulting PDF data for three time-of-flight neutron powder diffractometers has been recently reported [17]. Powder diffraction data with higher reciprocal space resolution led to PDF data with much lower damping.

On the other hand, nowadays, it is easy to find a lot of different kinds of materials that combine amorphous/nanocrystalline/crystalline phases, for instance, cements, pharmaceuticals, alloys, glass-ceramics, etc. The characterization of the nanocrystalline/amorphous materials in mixtures with large amounts of crystalline phases is very challenging, which is the case for cement hydrated samples.

Portland cement (PC) is one of the most manufactured products in the world, as it is a main component of the construction industry [18]. Its main hydration reaction consists of the dissolution of tricalcium silicate, Ca_3_SiO_5_, and then, the crystallization of portlandite, Ca(OH)_2_ and the precipitation of a nanocrystalline calcium–silicate–hydrate (C–S–H) gel [19,20,21], according to the following reaction:Ca_3_SiO_5_ + 5.2H_2_O → crystalline: 1.2Ca(OH)_2_ + nanocrystalline/gel: (CaO)_1.8_SiO_2_(H_2_O)_4_(2)

Moreover, there are some alternatives to PC which can be considered as environmentally friendly materials because they lead to a decrease in CO_2_ emissions, for instance, calcium sulfoaluminate (CSA) cements [22]. Its main phase is ye’elimite, Ca_4_Al_6_O_12_SO_4_, and its dissolution and hydration reaction, in the presence of a sulfate source, leads to the crystallization of ettringite and the precipitation of nanocrystalline gibbsite [13,23], according to the following reaction:Ca_4_Al_6_O_12_SO_4_ + 2CaSO_4_·0.5H_2_O + 37H_2_O→ crystalline: Ca_6_Al_2_(OH)_12_(SO_4_)_3_(H_2_O)_26_ + nanocrystalline: 4Al(OH)_3_(3)

The PDF methodology, using high-energy synchrotron radiation, has already been applied to the characterization of different cement hydrated samples [24,25]. For instance, the local structure of the C–S–H gel has been deeply studied. It was found that the C–S–H gel has a nanocrystalline nature with a local atomic ordering close to 4 nm [26,27,28]. Moreover, the C–S–H gel directly obtained from the hydration of tricalcium silicate has been very recently reported to contain two components: a nanocrystalline defective clinotobermorite, Ca_11_Si_9_O_28_(OH)_2_·8.5H_2_O, and an amorphous component which was described as isolated monolayers of calcium hydroxide [21,29]. In addition, the nanocrystalline gibbsite resulting from ye’elimite-containing pastes have also been studied by the PDF approach. It was found that thus aluminum hydroxide had a gibbsite-type local structure with a particle size close to 3 nm [13].

The main objective of this work is to compare different experimental configurations for synchrotron total scattering to determine the implications in the derived PDF datasets and the consequences for the outputs of the analyses. The final goal is to obtain the best possible characterization including quantitative phase analysis for mixtures containing nanocrystalline and crystalline phases. In order to do so, PDF data have been collected using three experimental configurations at beamlines: ID151A and ID22 of european synchrotron radiation facility (ESRF) synchrotron and materials science and powder diffraction (MSPD) of ALBA synchrotron. Initially, a (very simple) single phase, highly crystalline nickel, was used as standard. Subsequently, three sets of real samples related to cements were analyzed. The robustness of the procedures and quality of the data based on different criteria are discussed.

## 2. Materials and Methods

### 2.1. Samples Preparation

In this study, a crystalline standard and different research samples, related to the cement field, were used. They are described next.(a)Crystalline nickel was used as a standard. This was a commercial powder sample (99.9%, 3–7 μm of average particle size) purchased from Strem Chemical INC.(b)Stoichiometric triclinic tricalcium silicate, Ca_3_SiO_5_, was purchased from Mineral Research Processing (M.R.PRO) and it was a single crystalline complex phase.(c)Doped dicalcium silicate, Ca_2_Si_0.972_Al_0.028_O_3.986_□_0.014_, was synthesized as described in Reference [30] to stabilized the monoclinic structure at room temperature. This sample, labeled hereafter as Ca_2_SiO_4_, was composed mainly by β-Ca_2_SiO_4_ but it contains about 10 wt.% of γ-Ca_2_SiO_4_, see below.(d)Stoichiometric ye’elimite, Ca_4_Al_6_O_12_SO_4_, mixed with bassanite, CaSO_4_·0.5H_2_O, was hydrated at a water-to-solid (w/s) mass ratio of 1.2 for 14 days. The resulting sample was already reported in a previous publication [13]. The sample is a mixture of crystalline ettringite and a nanocrystalline gibbsite.(e)Monoclinic tricalcium silicate was purchased from Mineral Research Processing (M.R.PRO). This commercial sample was already described in Reference [13]. As reported, the sample was hydrated with w/c ratio of 0.8 for 34 days. The sample was a mixture of crystalline portlandite and a nanocrystalline C–S–H gel containing a small fraction of unreacted tricalcium silicate.

The hydrated samples, (d) and (e) were prepared as previously reported [13]. The pastes were prepared in Teflon® cylinder shaped molds for 24 h and then stored in demineralized water. At the selected age, samples were taken out and the excess of water were removed by washing the sample twice with isopropanol and once with ether [13].

All the samples were introduced into glass capillaries of 0.7 mm of diameter. The capillaries were sealed with grease to avoid any water loss for the hydrated mixtures.

### 2.2. Total Scattering Synchrotron X-ray Powder Diffraction (SXRPD)

The Scattering Synchrotron X-ray Powder Diffraction (SXRPD) data were collected at three different beamlines in two synchrotron facilities (ESRF and ALBA):(a)Beamline ID15A at ESRF, European Synchrotron Radiation Facility, (Grenoble, France): The white beam was monochromatized by a liquid nitrogen cooled double Laue Si monochromator and the wavelength was 0.18972 Å (65.35 KeV). The beam size was 100 μm horizontal and 100 μm vertical to avoid beam size-dependent Q-broadening. The SXRPD data were collected in transmission geometry using a Pilatus3 X CdTe 2M hybrid photon counting 2D detector with 172 μm of pixel size located in a motorized stage. The detector was off-centered with respect to the incident beam and located close to the sample in order to acquire the largest possible range of the pair distribution function G(r). The detector distance was calibrated with CeO_2_ and it was found to be 181.592 mm which allows reaching a Q_max ins_ close to 30 Å^−1^. The data acquisition time was 8 s per individual image. For the hydrated pastes, and in order to increase the statistics of the final data sets, eight images of 8 s were collected and merged with local software.(b)Beamline ID22 at ESRF: The white beam was monochromatized by a channel cut Si 111 monochromator and the wavelength was 0.20665 Å (60 KeV) [31]. The data were collected in transmission geometry using a Perkin Elmer XRD 1611CP3 2D detector with a 100 μm of pixel size. The beam arriving at the entrance to the refracting lenses was 0.5 × 0.5 mm^2^ and the focused beamsize at the 2D detector was 150 μm horizontal and 50 μm vertical. This detector was installed on a motorized linear translation stage that allows for variation of the sample-detector distance between 2500 and 380 mm. By using this configuration, it was possible to reach a maximum momentum transfer, Q_max ins_, of 27 Å^−1^. The total acquisition time per single pattern was 27 s. Moreover, other total acquisitions times were also tested, 67, 400, and 1004 s.(c)Beamline MSPD at ALBA synchrotron (Barcelona, Spain): The white beam was monochromatized by a double-crystal Si (111) monochromator and the wavelength was 0.41236(1) Å (30.067 keV) [32]. The data were collected in transmission geometry using a six modules MYTHEN 1D detector (strip-pitch: 50 µm, radius: 550 mm, covered angular range: 40°). The shortest usable wavelength was dictated by the efficiency of the Si-detector as higher energy X-ray, which would allow a larger Q-range to be probed, would yield poorer signal at a fraction of the scattered photons would not be measured by this detector. The beamsize was 3 mm horizontal and 1 mm vertical and it was focused vertically onto the detector. The data acquisition time was 37 min per each individual pattern. Three patterns were collected for the anhydrous samples and five for the hydrated pastes in order to improve the signal-to-noise ratio in the large recorded angular range, 1° to 120° (2θ). The maximum momentum transfer obtained was Q_max ins_ of 25 Å^−1^.

For all the experiments, the same size of the glass capillaries was used, 0.7 mm of diameter. The capillaries were always rotated during data collection which yield smooth Debye–Scherrer rings which are easy to integrate. Furthermore, rotating the capillaries for 1D detectors is essential to improve diffracting particle statistics and hence to have reliable intensity values.

### 2.3. PDF Data Analysis

The PDF experimental data were obtained using PDFgetX3 [33] with Q_max_ = 24 Å^−1^ for the three experimental configurations studied. Moreover, for ID15A and ID22 datasets, the PDF profiles were also obtained using Q_max_ of 29 and 26 Å^−1^, respectively. Structural information and quantitative phase analysis results were obtained from the PDF data by using the PDFGui software [34]. For the nickel-standard the following parameters were optimized/determined: scale factor, unit cell parameter, delta2 parameter (low-r correlated motion peak sharpening factor) [35,36], the instrumental parameters Q_damp_ and Q_broad_, and ADPs (Atomic Displacement Parameters).

For the others samples, the final global optimized parameters were: scale factors, unit cell parameters, delta2 parameter (for the fits containing the low r-regions) and ADPs. In some cases, as described in the Results section, atomic positional coordinates were also varied. Moreover, the spdiameter parameter was also adjusted to fit the nanocrystalline component in the ye’elimite-bassanite sample. It is important to point out that we do not report errors for the refined parameters as the standard deviations are not properly calculated by PDFgui. This is because the standard data reduction obtained by PDFgetX3 [33] does not include the dG column, standard deviations of G, which is needed to properly calculate the errors.

### 2.4. Laboratory X-ray Powder Diffraction (LXRPD) and Rietveld Analysis

The LXRPD data for the cement samples were collected on a D8 ADVANCE (Bruker AXS) diffractometer equipped with a Johansson monochromator (at SCAI—Universidad de Malaga), using strictly monochromatic Mo-Kα_1_ radiation, λ = 0.7093 Å, in transmission geometry (θ/θ). The LXRPD data for nickel standard was collected on a PANalytical X’Pert Pro MPD automated diffractometer (PANalytical) equipped with a Ge(111) primary monochromator, using strictly monochromatic CuKα_1_ radiation (λ = 1.54059 Å) and an X’Celerator detector. Rietveld analyses were performed using the GSAS suite of programs and the EXPGUI graphic interface [37]. Final global optimized parameters were: background coefficients, zero-shift error, cell parameters, and peak shape parameters using a pseudo-Voigt function. Moreover, the ADPs were also optimized for the nickel standard.

## 3. Results and Discussion

### 3.1. Nickel-Standard

Crystalline nickel was selected to compare the PDF data collected at different beamlines, with different detector systems, as described in the Materials and Method section. The plots of the raw PDF data measured in the three experimental configurations are displayed in Appendix A.

A preliminary study was dedicated to the overall acquisition time for this standard. (i) For the total scattering data collected at ID22, several data acquisition times were tested: 27, 67, 400, and 1004 s, see Appendix A. All the PDF fits led to the same results with very similar R_W_ values as it is shown in Appendix A. Consequently, the lowest data acquisition time of 27 s was selected for this configuration and sample. (ii) For the total scattering data collected at MSPD, two different data collection times were also tested: 37 min and 74 min (two cycles of 37 min), see Appendix A. As it is shown in Appendix A, the differences in the results of the PDF fits obtained were negligible. So, the selected data acquisition time was 37 min. (iii) For ID15A data, only one overall acquisition time, 8 s, was tested which yielded high quality PDF data.

The PDF data for crystalline Ni, collected at the three beamlines with the acquisition times discussed just above, were fitted up to 150 Å. A summary of the obtained results are shown in Table 1. The unit cell and ADP values obtained from the Rietveld fit are also included in Table 1 for the sake of comparison. Moreover, Appendix A shows the Rietveld plot for the crystalline nickel. For the ID15A data, patterns processed up to two different Q_max_ values, 24 and 29 Å^−1^, were analyzed, with Figure 1a showing the PDF fit using the Q_max_ = 24 Å^−1^ dataset. A similar study was performed for the ID22 data, in this case Q_max_ was 24 and 26 Å^−1^. Figure 1b shows the PDF fit using the Q_max_ = 24 Å^−1^ dataset. For the MSPD data, only one dataset, Q_max_ = 24 Å^−1^, was recorded and the PDF fit is shown in Figure 1c. Finally, Q_damp_ parameter for MSPD dataset is much smaller than that of ID15A, with the value for ID22 being intermediate, see Table 1. As previously discussed [17], this difference in damping’s is a consequence of the different resolutions of the data in Q-space. Appendix A displays the three different diffraction datasets to highlight this.

Figure 2 displays the PDF fits for the nickel sample in the 1.35–10 Å region for the analyses described just above, 1.35–150 Å analyzed r-range. As expected, the PDF data collected at ID15A, processed up to Q_max_ = 29 Å^−1^, see Figure 2b, showed less intense ripples below 4 Å that the corresponding dataset processed up to Q_max_ = 24 Å^−1^, see Figure 2a. This is highlighted in Figure 2 by blue arrows. The flattening of the ripples for the ID22 data is much less pronounced. PDF data processed up to Q_max_ = 24 Å^−1^, Figure 2c, show very similar ripples to data processed up to Q_max_ = 26 Å^−1^, Figure 2d. PDF data processed up to Q_max_ = 24 Å^−1^ for MSPD, see Figure 2e, shows the largest ripples which extend slightly beyond 4 Å. The raw intrinsic high resolution in Q (widest possible dataset) is very important as data collected at ID15A with Q_max ins_ = 30 Å^−1^ but processed up to Q_max_ = 24 Å^−1^, see Figure 2a, shows much less intense ripples than the pattern collected at MSPD with Q_max ins_ = 25 Å^−1^ and processed up to Q_max_ = 24 Å^−1^, see Figure 2e.

The role of the Q_max_ in the real space resolution has been previously treated [38] where larger values of Q_max_ led to better resolution is real space. As it is shown in Figure 2, highlighted by red arrows, the MSPD data yielded narrower interatomic peaks compared to those obtained from data collected at ID15A and ID22. This result is important for quantitative phase of materials containing crystalline and nanocrystalline (and even amorphous) components as less overlapping means more information. This observation, narrower interatomic peaks for MSPD data than for ID22 and ID15A data may be in disagreement with the Q_broad_ calculated for the three beamline datasets, see Table 1 as the Q_broad_ parameter of the MSPD and ID22 analyses were very close. For this reason, a deeper study has been carried out.

In order to quantify the resolution of each dataset for the interatomic peaks, three different Ni–Ni interatomic distance peaks were selected. The full width at half maximum FWHM values for these peaks are given in Table 2 and fits are displayed in Appendix A. It is clear that the interatomic peaks for the MSPD PDF pattern are ~15% narrower than those obtained at ID15A and ID22 in this r-range. Furthermore, anisotropic interatomic distance peak broadening is also measured as the Ni–Ni peaks for 3.63 Å are clearly broader than those corresponding to 2.49 and 6.58 Å.

In order to understand the observation of similar Q_broad_ values for the MSPD and ID22 datasets (r-range: 1.35–150 Å) an additional study was carried out. The three Ni PDF datasets were analyzed in three different r-regions (1.35–30, 30–50, and 50–100 Å). Table 3 reports the Q_broad_ parameters obtained in these nine fits and the corresponding plots are displayed in Appendix A. For the 1.35–30 Å r-region, indeed the Q_broad_ value obtained for MSPD dataset was much smaller than that obtained for ID22, in agreement with the observation reported above. Appendix A shows the PDF fits in this low r-region for ID15A, ID22 and MSPD datasets, respectively. A similar study was also carried out for the other two r-regions. Table 3 shows that the Q_broad_ parameter in the 30–50 Å r-range was very similar for the ID22 and MSPD datasets. Hence, the Q_broad_ parameter for MSPD, which was very small in the 1.35–30 Å range, slightly increased in the 30–50 Å range. For ID15A, which showed the broadest peaks, the change in Q_broad_ parameter in the studied regions was negligible. The corresponding plots are displayed in Appendix A. Finally, for the 50 to 100 Å r-region, see Appendix A, the results showed in Table 3 indicates that the width of the Ni peaks did not change significantly.

### 3.2. Cement Samples

Three additional set of samples, related with building materials, have been measured in two experimental configurations to compare the obtained results. On the one hand, ID15A configuration allows to access the highest resolution in reciprocal space, Q_max ins_ = 30 Å^−1^, with very short acquisition times that allow fast kinetics studies to be carried out. On the other hand, MSPD configuration allows to obtain the highest resolution in the interatomic distances (real) space, which may be useful for the quantitative phase analysis of mixtures containing crystalline and nanocrystalline components.

#### 3.2.1. Anhydrous Triclinic Tricalcium Silicate and Dicalcium Silicate Samples

Two anhydrous calcium silicates samples, triclinic Ca_3_SiO_5_ and monoclinic Ca_2_SiO_4_, were selected for the sake of comparison of the results obtained from ID15A and MSPD datasets. The PDF fits for these samples are also important as they remain partly unreacted along the hydration process and their contribution to the (overall) measured PDF signal must be properly modelled.

Rietveld quantitative phase analysis showed that t-Ca_3_SiO_5_ was a single crystalline phase, see Appendix A. The Rietveld analysis for the monoclinic dicalcium silicate sample, see Appendix A, showed that it is composed of 91.6(1) wt.% of β-Ca_2_SiO_4_ and 8.4(3) wt.% γ-Ca_2_SiO_4_.

The PDF fits for t-Ca_3_SiO_5_ were performed from 1.4 to 50 Å and the final *R*_W_ values were 11.1% and 10.8% for ID15A and MSPD datasets, respectively. Appendix A display the final PDF fits in the whole analyzed r-range. Moreover, Figure 3a,b highlight the low r-region, 1.4–10 Å, plots for these fits. The PDF analyses for monoclinic Ca_2_SiO_4_ were also performed from 1.4 to 50 Å and the final *R*_W_ values were 15.7% and 17.5% for ID15A and MSPD datasets, respectively. For γ-Ca_2_SiO_4_ phase only the scale factor and the unit cell parameters were refined. Appendix A show the final PDF fits. Figure 3c,d highlight the low r-region, 1.4–10 Å, plots for these fits. Appendix A reports selected results of the four fits.

The visual inspection of Figure 3 show that the interatomic distances peaks for t-Ca_3_SiO_5_ were broader than the corresponding ones for Ca_2_SiO_4_. This was expected as the crystal structure of triclinic calcium silicate is much more complex than that of dicalcium silicate. The t-Ca_3_SiO_5_ has 29, 9 and 45 crystallographically independent Ca, Si, and O atoms, respectively. Therefore, there are many more (slightly) different interatomic distances that result in the broadening of the t-Ca_3_SiO_5_ PDF peaks, when compared to those of Ca_2_SiO_4_. Furthermore, for the same sample the MSPD PDF patterns displayed slightly narrower interatomic distance peaks in agreement with the study shown above for the standard. This result reflects the smaller contribution from the MSPD instrument to the overall resolution in real space.

On the other hand, monoclinic Ca_2_SiO_4_ contains a second crystalline phase, determined by well-established Rietveld methodology, which may serve to test the accuracy of the quantitative phase analyses by PDF. The ID15A PDF data fit yielded 89.9 and 10.1 wt.% for β-Ca_2_SiO_4_ and γ-Ca_2_SiO_4_, respectively. The MSPD PDF data fit gave 91.7 and 8.3 wt.% for β-Ca_2_SiO_4_ and γ-Ca_2_SiO_4_, respectively. Both results are good and the MSPD PDF result is identical to the laboratory Rietveld quantitative phase analysis within the errors, 8.4(3) wt.% γ-Ca_2_SiO_4_.

#### 3.2.2. Ye’elimite with Bassanite Hydrated Paste

Ye’elimite hydrates in the presence of bassanite according to the chemical reaction given in Equation (3). The reaction products are crystalline ettringite and nanocrystalline gibbsite. The analysis of PDF data collected at MSPD for a ye’elimite with bassanite hydrated paste has been recently reported [13]. Here, new PDF data has been collected at ID15A for exactly the same sample. Moreover, the PDF fits have been carried out using the strategy given in Reference [13].

The ID15A PDF fit for the crystalline phase, ettringite, was performed in the 30 to 50 Å r-region according to the previous publication [13] and it is shown in Appendix A. Then, all the parameters for ettringite were kept fixed and a second refinement was performed in the low r-region.

In the previous work [13], the PDF fit for crystalline ettringite and nanocrystalline gibbsite was performed between 1.6 and 35 Å. This choice was dictated by the ripples of the PDF data below 1.6 Å, which did not make possible the proper fit of the first interatomic peak, S–O, at a distance of ~1.50 Å. The ID15A PDF pattern allows data processing up to Q_max_ = 29 Å^−1^, and so, the ripples in the low r-region were minimized, as expected and demonstrated for the Ni standard, see Figure 2b. For this reason, the first attempt was to perform the PDF fit for this sample, ID15A data, from 1.35 to 35 Å, see Figure 4a. Figure 4b just highlights an enlarged view of the low r-range region. The attempts to fit the first S–O peak led to a poor PDF fit with high Rw value, 29.1% and the peak located at ~2.4 Å (Ca–O interatomic distance) could not be properly fitted. Selected results for this fit are reported in Table 4. Subsequently, a similar refinement was carried out but for the region 1.6 to 35 Å, resulting in a very good fit, Rw = 23.7%, see Figure 4c,d. The results for the MSPD PDF data for this sample are also shown in Table 4 and the plots in Figure 4e,f.

The results shown in Table 4 for the 1.6–35 Å region for ID15A and MSPD datasets were very similar. However, it must be noted that the *R*_W_ value for the MSPD fit is significantly higher. On the one hand, the real space resolution of the MSPD PDF data was larger as evidenced by the green arrows in Figure 4. The peak at ~2.9 Å of nanogibsite, Al–Al interatomic distance, was much less overlapped with the peak at ~2.4 Å, Ca–O bond distance, for the MSPD dataset. This is again an evidence of the higher resolution in real space of the MSPD data. On the other hand, there are (small) features in the PDF data which are not properly fitted. This is highlighted in Figure 4, for one case, with a black arrow. We speculate that these features are due to an additional nanocrystalline phase which would explain the differences between the measured crystalline ettringite and nanocrystalline gibbsite contents and the expected values from chemical Equation (3), see Table 4. The Rietveld plot, using the MSPD data, excludes the presence of crystalline AFm, see Appendix A. More research is needed to determine if these features arise from amorphous/nanocrystalline ettringite or from the alternative hydration reaction, see Equation (4), which may yield amorphous/nanocrystalline monosulfate, Ca_4_Al_2_O_6_(SO_4_)·14H_2_O [23,39]. Finally, we noted that unaccounted features are more evident at higher real space resolution, MSPD data, which would explain the larger disagreement *R*_W_ value for this fit when compared to the similar fit for ID15A data, see Table 4. By using an internal standard, the Rietveld-derived quantitative phase analysis reported in Reference [13] showed that this sample contained 64 wt.% of crystalline ettringite and 36 wt.% of overall amorphous content.

Ca_4_Al_6_O_12_(SO_4_) + 18H_2_O → Ca_4_Al_2_(OH)_12_(SO_4_)·6H_2_O + 4Al(OH)_3_(4)

#### 3.2.3. Monoclinic Tricalcium Silicate Hydrated Paste

For the monoclinic Ca_3_SiO_5_ hydrated paste, MSPD PDF data were already collected and reported [13]. Here, new PDF data were collected at ID15A for the same sample. The ID15A PDF data have been analyzed using the previous reported strategy [13].

The MSPD PDF fits already published are displayed in Figure 5 (right panels) in the different r-ranges. The high r-region, 40–70 Å, was analyzed to obtain the contributions mainly due to crystalline phases: Ca(OH)_2_, unreacted Ca_3_SiO_5_ and CaCO_3_-calcite as impurity formed by carbonation of portlandite, see Figure 5b. The ID15A PDF data analysis was carried out with the same approach and the plot for the 40–70 Å r-region is displayed in Figure 5a. The intensities for the ID15A dataset decreased more pronouncedly at large interatomic distances, above 55–60 Å. This is reflected in the larger Q_damp_ parameter obtained, 0.0221 Å^−1^, when compared to the corresponding value for the MSPD analysis, Q_damp_ = 0.0039 Å^−1^.

All the previous parameters were kept fixed and the contribution from nanocrystalline C–S–H gel was studied in the intermediate r-region, from 10 to 25 Å, see Figure 5c,d. Moreover, the phase analysis can also be compared with that already published, see Appendix A. Finally, the region from 2 to 10 Å was also analyzed and the results are shown in Figure 5e,f, although they were not thoroughly modeled in this work. As it was previously reported [21,29], the PDF misfits at low interatomic distances very likely correspond to the theoretical PDF trace of isolated monolayers of Ca(OH)_2_.

## 4. Conclusions

Here, we report synchrotron PDF data collected at three beamlines in different experimental configurations and X-ray detectors. We have shown that 2D detectors collected very good data in less than 10–20 s but 1D detectors need about 30 min. We have used highly crystalline Ni single phase to characterize the resolution in both momentum transfer and real spaces. As expected, total scattering data collected with large Q values, 29 Å^−1^ resulted in PDF data with negligible ripples. However, data acquired up to Q_max_ = 24 Å^−1^ gave ripples extended to 4 Å (for this very crystalline sample). Interestingly, the real space resolution was higher for the dataset taken with the 1D detector and λ = 0.41 Å. It is highlighted that the highest possible real-space resolution could be important for the analysis of samples containing a mixture of crystalline and amorphous/nanocrystalline phases.

Subsequently, three sets of cement-related samples have been analyzed. The widths of the interatomic peaks for anhydrous triclinic Ca_3_SiO_5_ and Ca_2_SiO_4_ samples have been characterized for two different experimental configurations. For a sample resulting from the hydration of ye’elimite and bassanite, which contained at least crystalline ettringite and nanocrystalline gibbsite, the high resolution in real space from MSPD allowed to identify an interatomic peak which arises from an unknown component. This experimental finding shows the importance of high resolution in complex mixtures. Finally, a paste resulting from the hydration of monoclinic Ca_3_SiO_5_ was also measured at two beamlines and the analyses of the profiles are reported.

## Figures and Tables

**Figure 1 materials-12-01347-f001:**
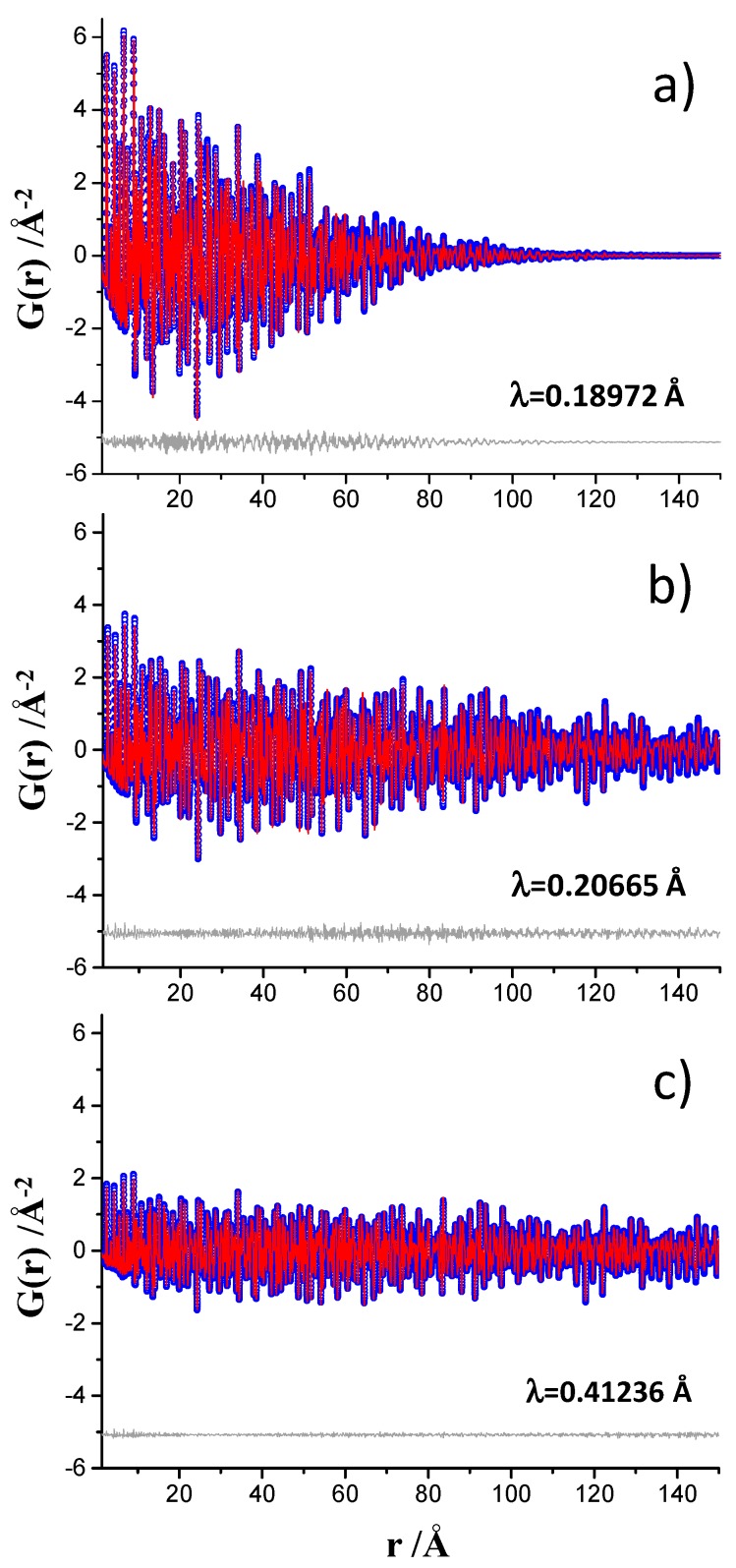
Experimental (blue circles) and fitted (red solid line) PDF fit profiles for Ni standard in the 1.35 to 150 Å r-range, (**a**) data collected at ID15A, (**b**) data collected at ID22, and (**c**) data collected at MSPD. Difference curves are shown as grey lines. Q_max_ = 24 Å^−1^.

**Figure 2 materials-12-01347-f002:**
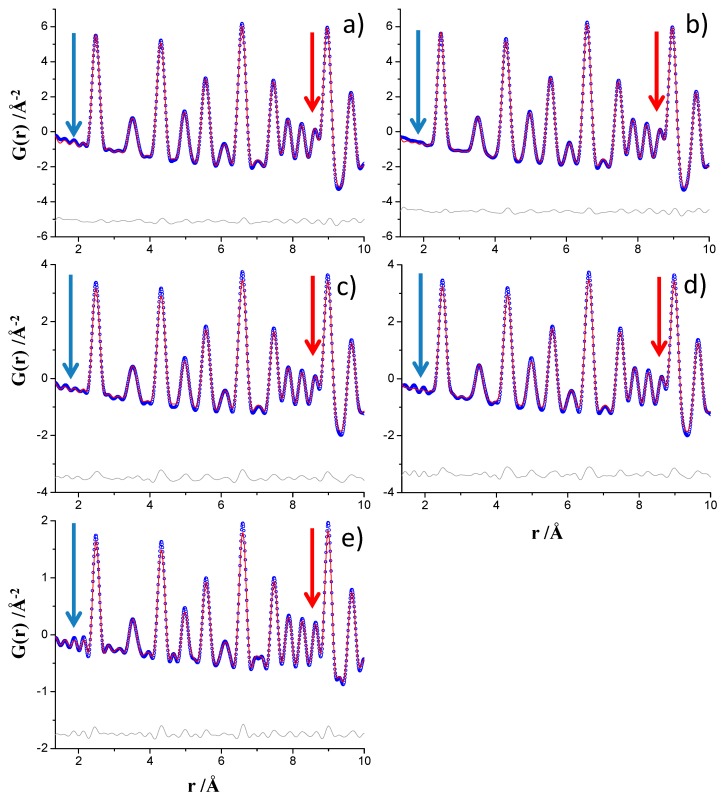
Experimental (blue circles) and fitted (red solid line) PDF fit profiles for Ni standard in the 1.35 to 10 Å r-range, (**a**) ID15A data at Q_max_ = 24 Å^−1^, (**b**) ID15A data at Q_max_ = 29 Å^−1^, (**c**) ID22 data at Q_max_ = 24 Å^−1^, (**d**) ID22 data at Q_max_ = 26 Å^−1^ and (**e**) MSPD data at Q_max_ = 24 Å^−1^. Difference curves are shown as grey lines. The blue arrows highlight the ripples. The red arrows highlight the lower overlapping due to narrowing of the interatomic pair peaks.

**Figure 3 materials-12-01347-f003:**
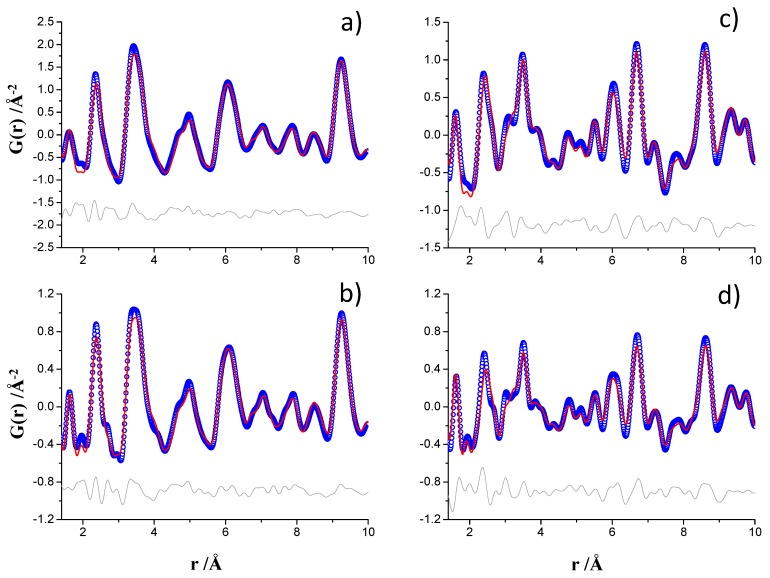
Experimental (blue circles) and fitted (red solid line) PDF fit profiles displaying the 1.4–10 Å r-range, Q_max_ = 24 Å^−1^, (**a**) ID15A data for triclinic Ca_3_SiO_5_, (**b**) MSPD data for triclinic Ca_3_SiO_5_, (**c**) ID15A data for monoclinic Ca_2_SiO_4_, and (**d**) MSPD data for monoclinic Ca_2_SiO_4_. Difference curves are shown as grey lines.

**Figure 4 materials-12-01347-f004:**
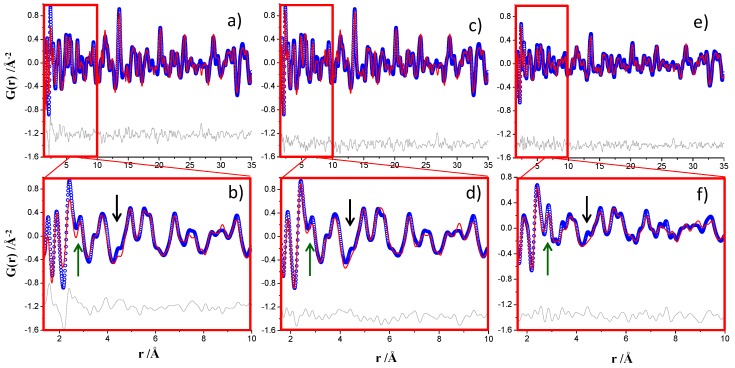
Experimental (blue circles) and fitted (red solid line) PDF fit profiles for ye’elimite with bassanite hydrated paste with w/s = 1.20 and T = RT for 14 days (**a**) ID15A data, 1.35–35 Å r-range, Q_max_ = 29 Å^−1^, (**b**) inset of the PDF fit showed in (**a**) highlighting the key 1.35–10 Å r-range, (**c**) ID15A data, 1.60–35 Å r-range, (**d**) inset of the PDF fit showed in (**c**) highlighting the 1.60–10 Å r-range, (**e**) MSPD data, 1.60–35 Å r-range, Q_max_ = 24 Å^−1^, [13] and (**f**) inset of the PDF fit showed in (**e**) highlighting the 1.60–10 Å r-range. Difference curves are shown as grey lines. The green arrows highlight the (slightly) better resolution, real space, of the MSPD data as there is less overlapping between the interatomic peaks. The black arrows highlight unaccounted features which are more conspicuous in the MSPD dataset, likely due to its higher resolution.

**Figure 5 materials-12-01347-f005:**
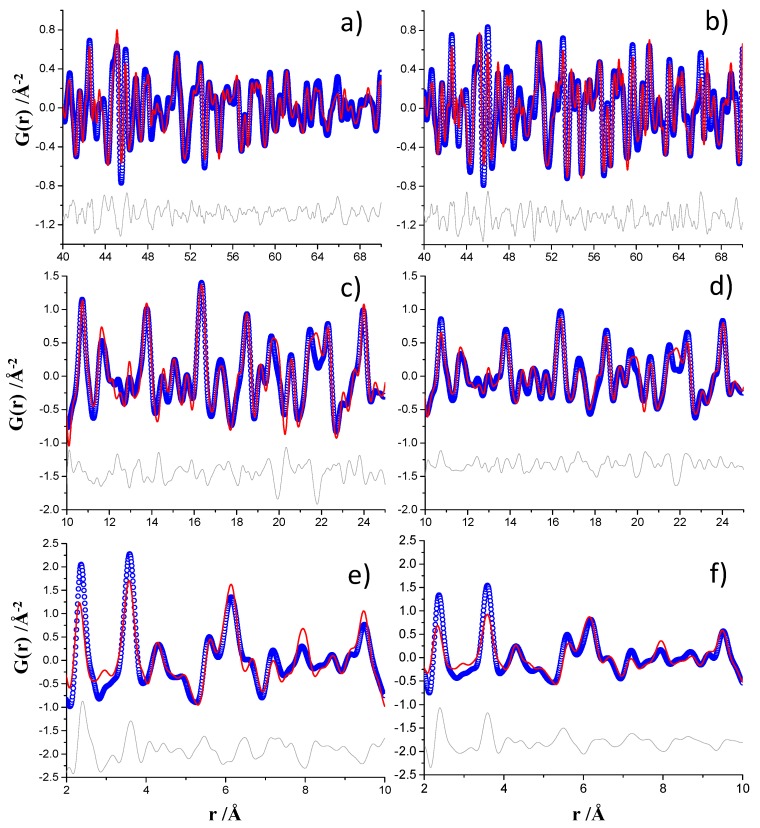
Experimental (blue circles) and fitted (red solid line) PDF fit profiles for monoclinic Ca_3_SiO_5_ hydrated paste, Q_max_ = 24 Å^−1^, (**a**) ID15A data for 40–70 Å r-range, (**b**) MSPD data for 40–70 Å r-range [13], (**c**) ID15A data for 10–25 Å r-range, (**d**) MSPD data for 10–25 Å r-range [13], (**e**) ID15A data for 2–10 Å r-range and (**f**) MSPD data for 2–10 Å r-range [13]. Difference curves are shown as grey lines.

**Table 1 materials-12-01347-t001:** Selected experimental details and instrumental parameters obtained from the pair distribution function (PDF) fits of the nickel standard in the 1.35–150 Å *r*-range for the studied configuration.

Beamline	ID15A		ID22		MSPD (Materials Science and Powder Diffraction)
λ (Å)	0.1897		0.2067		0.4124
Q_max ins_ (Å^−1^)	30		27		25
Detector	Pilatus3 X CdTe 2M 2D		Perkin Elmer XRD 1611CP3 2D		6 modules MYTHEN 1D
Acq. time (s)	8		27		2220
Q_max_ (Å^−1^)	24	29	24	26	24
Q_damp_ (Å^−1^)	0.0221	0.0221	0.0094	0.0094	0.0039
Q_broad_ (Å^−1^)	0.0143	0.0142	0.0084	0.0084	0.0085
Unit cell (Å)^#^	3.5178	3.5178	3.5245	3.5245	3.5263
ADP (Å^2^)^#^	0.0059	0.0060	0.0061	0.0061	0.0042
delta2 (Å^2^)	1.81	1.79	1.84	1.90	2.62
R_W_ (%)	9.0	9.0	9.5	9.6	5.8

^#^ Unit cell value for this crystalline Ni sample determined by Rietveld fit of CuKα_1_ LXRPD data was 3.52666(2) Å. The ADP value was 0.0122(2) Å^2^.

**Table 2 materials-12-01347-t002:** Full width at half maximum (FWHM) values for three analyzed Ni–Ni peaks obtained from the PDF patterns (Q_max_ = 24 Å^−1^) for each experimental configuration.

Beamline	ID15A	ID22	MSPD
	FWHM (Å)		
Ni–Ni at 2.49 Å	0.237	0.236	0.208
Ni–Ni at 3.63 Å	0.279	0.308	0.240
Ni–Ni at 6.58 Å	0.253	0.254	0.222

**Table 3 materials-12-01347-t003:** *Q*_broad_ parameters obtained for the crystalline Ni PDF fits in selected r-ranges for each experimental configuration.

Beamline		ID15A	ID22	MSPD
1.35–30 Å	Q_broad_ (Å^−1^)	0.0153	0.0065	0.0003
	R_W_ (%)	6.5	6.1	5.2
30–50 Å	Q_broad_ (Å^−1^)	0.0146	0.0094	0.0097
	R_W_ (%)	10.5	6.9	3.6
50–100 Å	Q_broad_ (Å^−1^)	0.0138	0.0089	0.0091
	R_W_ (%)	18.3	11.7	4.5

**Table 4 materials-12-01347-t004:** Quantitative phase analysis results for the ye’elimite with bassanite hydrated paste obtained from the PDF refinements. Spdiameter and R_W_ values are also included.

Beamline	ID15A 1.35–35 Å	ID15A 1.6–35 Å	MSPD [13] 1.6–35 Å	Expected According to (3) *
Ettringite (wt.%)	74.8	61.9	59.9	80.1
Nanocrystalline Al(OH)_3_ (wt.%)	25.2	38.1	40.1	19.9
Spdiameter (Å)	33	27	26	-
R_W_ (%)	29.1	23.7	28.4	-

* Theoretical weight percentages of expected hydration products according to Equation (3).

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
