# Peer review of "A Comparative Study of Experimental Configurations in Synchrotron Pair Distribution Function"

_materials, 2019, doi:10.3390/ma12081347_

Reviewer 1 Report

The paper "A comparative study of experimental configurations in synchrotron pair distribution function" compares, in its first part, three different beamline setups for collecting data for pair distribution function analysis.

The three beamline are ID15, ID22 (ESRF) and MSPD (ALBA). The authors report most of the experimental conditions, but not all of them.

In fact, for the beamline ID22 and MSPD they do not mention which was the beam size used. As this is one of the parameters that determines the resolution of the instrument (together with detector pixel size and sample-detector distance), and, as this is meant to be a comparative study to help possible users to choose which are the best conditions for PDF experiment, I would recommend the authors to complete these information.

Another parameter that the author should provide, still for the sake of comparison, is the flux at the sample.  Together with the time spent for collecting each set of data (already mentioned by the authors) this would complete the first part of the paper.

At line 218, the authors mention "' Conversely, the resolution in the interatomic peaks is not dependent of Qmax". This is not completely true, as the real space resolution is mainly dependent by pi/Qmax. I would ask the author to clarify this point. There are other evidences in the literature (see e.g. Bernasconi et al.

Total scattering experiments on glass and crystalline materials at the ESRF at the ID11 Beamline - 2015) that shows how the Qmax influences the resolution in real space.

The authors mention that "MSPD configuration allows  to obtain the highest resolution in the interatomic distances (real) space". As they have collected some data on ID22, have they tried to measure some samples in the high resolution mode? (with the multianalyser detector) . are there some data available for completing this comparison?

At line 249, what it the explanation why the Qbroad increases more dramatically for MSPD and stays flat for ID15? 

In the second part of the paper, the authors compare the result of some PDF analysis performed only on ID15 and MSPD on cement samples. This part is complete and all the fit the authors show are reasonable.

I would recommend the authors to better explains the issues raised above, mainly for the first part of the paper, and after that the editors to accept the paper after minor revisions

Author Response

The reply to reviewer 1 is attached in a PDF file

Reviewer 2 Report

The manuscript “materials-484389” reports a detail investigation on different experimental setups in synchrotron PDF studies. Authors provide an interesting and exhaustive series of studies and also the quantity of datasets are appreciable. Paper is written in appropriate way and can be accepted for publication after a few minor revisions outlined here below:

1)      To understand the quality of the refined data it is very important that the standard deviation is clarify for each reported parameters. Please, reported this data for each unit cell parameters refined and for the ADPs values.   

2)      In Table 4 and Table S4 the nomenclature "according to (3)" and "according to (2)", respectively, is not immediately understandable. I suggest to authors make the parenthesis explicit with a comment below the table.

3)      Please, check the references paying more attention. Below I reported the corrections:.

Line 444: delete  “Bhuvanesh, N. and Reibenspies”

Lines 447-448: I suggest to use this format “Egami, T. and  Billinge, S.J.L. Underneath the Bragg peaks: structural analysis of complex materials. Vol. 16. Newnes, 2012”.

Line 449: delete “Thorpe, M.F.” and check the reference (Billinge, S.J.L. and Thorpe, M. F. eds. Local structure from diffraction. Springer Science & Business Media, 2006. Is it correct this that I found?)

Line 452: replace “x” with “X”

Line 453: replace “;” with “,” and add the last page (5078-5088)

Line 454: replace “;” with “,” and add the last page (6464-6476)

Line 456: replace “;” with “,” and delete “IUCr”

Line 461: add “13(10), 4239-4244”

Line 466: add “1-17”

Line 467: delete “IUCr”

Line 470: delete “IUCr”

Line 484: rewrite as follow “Aranda, M. A. G., and Angeles G. De la Torre. "Sulfoaluminate cement." Eco-efficient concrete. Woodhead Publishing, 2013, 488-522”

Line 494: replace page with “195502-1-195502-4”

Author Response

The reply to reviewer 2 is attached in a PDF file
